# On the robustness of regressing tumor percentage as an explainable detector in histopathology whole-slide images

Marina D'Amato                    MARINA.DAMATO@RADBOUDUMC.NL
Maschenka Balkenhol              MASCHENKA.BALKENHOL@RADBOUDUMC.NL
Mart van Rijthoven               MART.VANRIJTHOVEN@RADBOUDUMC.NL
Jeroen van der Laak              JEROEN.VANDERLAAK@RADBOUDUMC.NL
Francesco Ciompi                 FRANCESCO.CIOMPI@RADBOUDUMC.NL
*Department of Pathology, Radboudumc, Nijmegen, Netherlands*

## Abstract

In recent years, Multiple Instance Learning (MIL) approaches have gained popularity to address the task of weakly-supervised tumor detection in whole-slide images (WSIs). However, standard MIL relies on classification methods for tumor detection that require negative control, i.e., *tumor-free* cases, which are challenging to obtain in real-world clinical scenarios, especially when considering surgical resection specimens. Inspired by recent work, in this paper we tackle tumor detection via a MIL-like weakly-supervised regression approach to predict the percentage of tumor present in WSIs, a clinically available target that allows to overcome the problem of need for manual annotations or presence of tumor-free slides. We characterize the quality of such a target by investigating its robustness in the presence of noise on regression percentages and provide explainability through attention maps. We test our approach on breast cancer data from primary tumor and lymph node metastases.

**Keywords:** Weakly-supervised learning, Explainability, Tumor detection, Histopathology

## 1. Introduction

Tumor detection in histopathology is a critical task of cancer diagnosis that can be partly automated with computer algorithms in several tissue types. However, training supervised methods that rely on pixel or patch-level annotations can be challenging due to the time-consuming nature of annotating histopathology images, which requires expertise from pathologists, especially when aiming at generic tumor detectors that can work across multiple types of tissue and cancers. As an alternative, weakly supervised methods via Multiple Instance Learning (MIL) have been proposed for binary classification problems (Ilse et al., 2018; Campanella et al.) using slide-level labels instead of manual annotations. One of the challenges of using binary classification methods is the need for a large dataset with both positive (tumor present) and negative (tumor absent) cases. However, in real-world clinical diagnostics, finding WSIs without tumor can be challenging, as most resected tissue specimens typically contain some degree of tumor.

Inspired by recent work (Lerousseau et al., 2021) where tumor percentages were used as weakly supervised targets to train a segmentation model, we propose a weakly-supervised regression-based approach for estimating the percentage of tumor in a WSI. This approach allows us to formulate a tumor detection pipeline without being hampered by the lack of negative cases. This task is also particularly relevant for clinicians, as percentage estimation is performed by pathologists for cases where molecular pathology is conducted, and therefore largely clinically available. However, these percentage estimations are often done visually and may be prone to noise. To ensure the robustness of our proposed framework, we conducted a target noise analysis to evaluate its performance under varying noise conditions.

Figure 1: Example of attention heatmaps for the different models.

## 2. Methods

We based our approach on CLAM (Lu et al., 2021), which we extended to a regression setting, while keeping its two main components: 1) patch embedding to 1024 features using a pretrained ResNet50, 2) aggregation of these features through attention-pooling.

**Data.** We used two datasets: the publicly available Camelyon16 (CAM16) dataset (Litjens et al., 2018), which includes 399 cases of lymph node images, and an internal dataset with 595 cases of triple negative breast cancer (TNBC) surgical resections. We generated the tissue masks using a tissue background segmentation algorithm (Bándi et al., 2019). From the non-background regions, we extracted non-overlapping 256x256 patches at a spatial resolution of 0.5 $\mu$m, which we embedded and passed to an attention model during training to assign attention scores. The models were trained using mean squared error (MSE) loss with Adam optimizer,

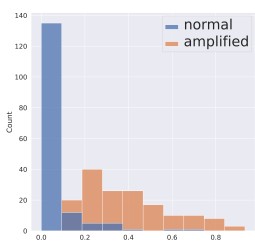

Figure 2: Distribution of labels before and after the amplification trick.

L2 weight decay of 1e-5 and a learning rate of 2e-4. We performed 5-fold cross-validation with a stratified train/validation/test split (65/15/20) based on the continuous target, while also rotating the test set to ensure coverage of the entire dataset in the evaluation process. To determine the tumor percentages used as regression targets, different strategies were employed for the two datasets. For CAM16, we used the existing manual tumor annotations, while for TNBC, segmentation maps generated using the HookNet algorithm (van Rijthoven et al., 2021) were utilized.

**Regression model and "target amplification trick".** The TNBC dataset, which only contains positive cases with tumor percentages ranging from 2% to 67%, showcases the potential of our approach to be trained without the requirement of tumor-free slides. In contrast, CAM16 includes both tumor and normal slides, with the percentages ranging from 0% to 70%. However, many CAM16 slides have a small percentage of tumor, with 91 slides having a percentage less than 1%. This narrow range of targets posed a challenge for our model to discriminate between tumor-free cases and slides with small lesions. To address this issue, we applied a *target amplification* "trick" by taking the fifth root of tumor percentages (Figure 2), effectively boosting lower values and making it easier for the model to discern subtle differences within this narrow range. We used two types of training approaches: (i) training and testing on each single dataset to analyze strengths and limitations of the models; (ii) cross-training on both datasets and testing on each individual one to examine generalization performance to different datasets with varying characteristics and distributional differences.

**On the effect of noisy targets.** We also assessed models robustness to noisy targets, mimicking visual estimation error in the clinic, by training models after injecting noise sampled from a uniform distribution, which decreased or increased the tumor percentage

| Training | Experiment | Pearson's r | MAE |
|---|---|---|---|
| TNBC | True percentages | 0.97 | 0.023 |
| TNBC | 10% noise | 0.94 | 0.033 |
| TNBC | 30% noise | 0.89 | 0.047 |
| TNBC | 50% noise | 0.76 | 0.075 |
| TNBC + CAM16 | True percentages | 0.96 | 0.025 |
| TNBC + CAM16 | 10% noise | 0.94 | 0.034 |
| TNBC + CAM16 | 30% noise | 0.86 | 0.052 |
| TNBC + CAM16 | 50% noise | 0.73 | 0.079 |

Table 1: Results obtained on TNBC dataset.

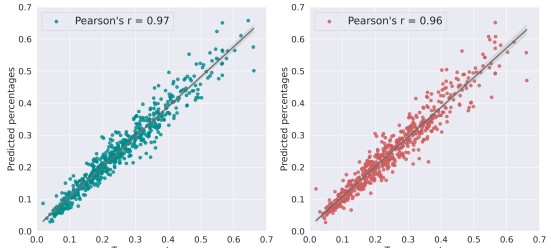

Figure 3: Scatter plots comparing the true and predicted percentages. Left: single training. Right: cross-training.

by 10%, 30% and 50% respectively. After each training with noisy tagrets, we evaluated the performance of the models on the test sets, which do not contain any noise.

## 3. Results and conclusion

Table 1 shows the results obtained on the TNBC dataset using the true percentages and the noisy targets for the two types of training respectively. Scatter plots in Figure 3 visually show the relationship between predicted and actual tumor percentages for the two types of training. ROC curves in Figure 4 demonstrate the performance of the model in performing the tumor detection task on CAM16, evaluated on all test sets from cross-validation and on the official test set separately. The predictions on the official test sets were generated by different models ensuring that the test sets data were not part of the training. Additionally, we can visualize the attention scores to gain insights into the patches that were crucial for the final prediction. Attention maps in Figure 1 reveal that the amplification trick enhances attention and improves visualization results, even when noisy training targets are used.

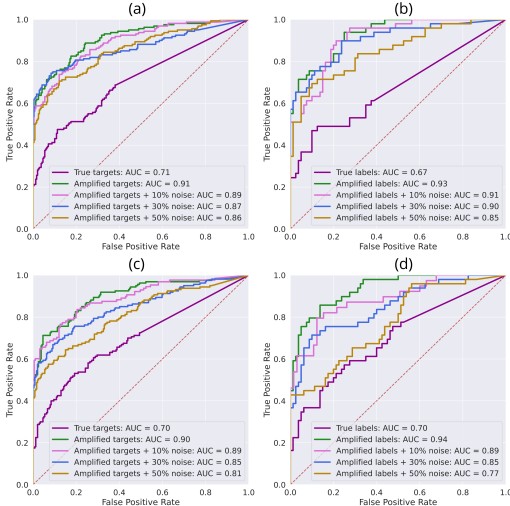

Figure 4: (a) CAM16 training - evaluation on all test sets. (b) CAM16 training - evaluation on official test set. (c) TNBC+CAM16 training - evaluation on all test sets. (d) TNBC+CAM16 training - evaluation on official test set.

In conclusion, our proposed approach, using CLAM in a regression setting, has shown promising results in addressing the task of tumor detection in WSIs without the need for manual annotations or tumor-free slides. Despite the expected decrease in performance when adding noise, the impact did not severely compromise the overall performance of the models. This indicates that our approach is potentially robust enough to handle noisy targets which may be encountered in real-world clinical scenarios and that tumor percentages can be used as a target for future weakly-supervised tumor detection. Furthermore, the cross-training experiments on both datasets demonstrated that combining different tissue types did not substantially impact the performance, showcasing the versatility and adaptability of our approach across different tissue types. In the future, we aim at achieving robust and accurate tumor detection across diverse cancer types by expanding the approach beyond breast cancer as explored in this study.

## 4. Acknowledgements

This project has received funding from the Innovative Medicines Initiative 2 Joint Undertaking under grant agreement No 945358. This Joint Undertaking receives support from the European Union's Horizon 2020 research and innovation program and EFPIA (www.imi.europe.eu).

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
