# OpenReview forum: "On the robustness of regressing tumor percentage as an explainable detector in histopathology whole-slide images"
_MIDL.io/2023/Short_Paper_Track — MIDL 2023 Short paper track Poster_

### Official Review · Reviewer_GBym · 2023-04-14
**good paper**

**Rating:** 7
**Confidence:** 4

**Review:**

The authors introduce a weakly-supervised regression-based approach for estimating the percentage of tumor in a WSI. The clinical focus is justified reasonably. The paper is very well written paper with interesting results that will generate healthy discussions during the conference.

---

### Official Review · Reviewer_L1cP · 2023-04-22
**Tumor percentage as an explainable detector in histopathology whole-slide images**

**Rating:** 6
**Confidence:** 5

**Review:**

This paper tackles tumor detection via a MIL-like weakly-supervised regression approach to predict the percentage of tumor present in WSIs, a clinically available target that allows to overcome the problem of need for manual annotations or presence of tumor-free slides. The experimental results were investigated on two datasets. Although it is interesting to see such a study on percentage prediction of tumors in WSI, how this facilitate the diagnosis and compare with existing studies with weakly supervised learning methods are missing.